# RETHINKING DATASET QUANTIZATION: EFFICIENT CORESET SELECTION VIA SEMANTICALLY-AWARE DATA AUGMENTATION

## ABSTRACT

Dataset quantization (DQ) is an innovative coreset selection method to choose representative subsets from large-scale datasets, such as ImageNet. Although DQ has made significant progress, it heavily relies on large pre-trained models (like MAEs), leading to substantial additional computational overhead. We first identify that removing this pre-trained MAE model degrades DQ's performance and increases the variance in model training. Where MAE plays a crucial role in introducing prior knowledge and implicit regularization into the training process. Second, we investigate a data augmentation scheme that can simulate the steps of pixel compression and reconstruction in DQ by simply using a randomly initialized ResNet model. This randomly initialized ResNet model can take advantage of the inductive bias of CNNs to locate the semantic object region and then replace the other region with other images. Therefore, we can use a random model or trained model in the early training stage to enhance semantic diversity while selecting important samples. We remove the module that contains the pre-trained MAE model and integrate the data augmentation scheme into the DQ pipeline, which formulates a new simple but efficient method, called DQ_v2. Our method achieves performance improvements across multiple datasets, such as ImageNette, CUB-200, and Food-101.

## 1 INTRODUCTION

Deep learning has become the golden standard for many computer vision and machine learning tasks (Dosovitskiy et al., 2021), which have seen rapid growth due to increasing model sizes and dataset volumes. However, training emerging deep models, e.g., vision transformers (ViTs) (Dosovitskiy et al., 2021), on large-scale datasets like ImageNet (Deng et al., 2009) and LIAON (Schuhmann et al., 2021) requires substantial computational resources, including high-performance GPUs, large memory capacity, and high-speed storage (Bartoldson et al., 2023). These requirements pose a significant barrier to entry for many researchers and practitioners, especially those in resource-constrained environments. Thus, how to efficiently train large-scale deep learning models with limited resources has become a common concern in both academia and industry.

Recent research has shown that large-scale datasets have many redundant and irrelevant samples (Xia et al., 2024; He et al., 2024), which can be compressed into a smaller representative subset without losing model performance. Thus, either coreset selection or dataset distillation, as crucial methods to address this issue, aims to choose or synthesize a representative subset from large-scale datasets to reduce computational complexity while maintaining model performance (Guo et al., 2022; Bartoldson et al., 2023). However, most existing methods face challenges in maintaining generalization and low scalability for larger datasets (Guo et al., 2022; Zhou et al., 2023). To this end, Dataset Quantization (DQ) is a recently proposed method that effectively addresses these challenges while maintaining high performance under all data keep ratios (Zhou et al., 2023; Zhao et al., 2024). We will quickly review the DQ method in the following section.

DQ is heavily based on various pre-trained models, that is, Masked Autoencoders (MAE) (He et al., 2022) and a pre-trained ResNet model (He et al., 2016). The ResNet model controls the dataset bin generation step, while the MAE model is used for image reconstruction in the pixel quantization and

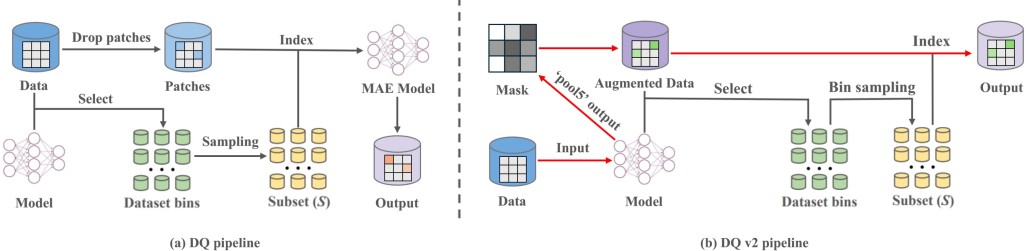

Figure 1: The overall pipeline of DQ and DQ_V2.

reconstruction step. These two steps are crucial for the DQ method to achieve high performance and stability. However, it is currently unclear whether these pre-trained models are crucial for the high performance of DQ methods. In addition, pre-trained models introduce additional computational overhead and inject prior knowledge into the selected subset, which may limit the generalization ability of trained models and affect the training stability.

Therefore, in this paper, we study the necessity of pre-trained models in the DQ method and propose a new method that achieves efficient coreset selection without relying on any pre-trained models. First, we observe that removing the MAE reconstruction step will significantly increase the variance in the trained model under different random seeds. That is, synthetic data has a beneficial effect on the training stability, which is crucial for the DQ method. Our preliminary analysis verify that the pre-trained MAE model is actually equivalent to a data augmentation method, which introduces prior knowledge and implicit regularization into the training process.

Therefore, inspired by the data augmentation method proposed by Tobias et al. (Cao & Wu, 2022), we propose a new data augmentation strategy that can replace the MAE model in the DQ method. This simple but effective data augmentation strategy leverages the inductive bias of random CNNs to effectively preserve the semantic structure while introducing beneficial variations. On the other hand, we can also use this model to split the dataset, which can be trained with several epochs from scratch similar to Paul et al. (2021). We integrate this semantically aware data augmentation strategy into the DQ framework without the MAE model and propose a new method, named Dataset Quantization V2 (DQ_V2). Figure 1 shows the pipeline of our DQ_V2. Intuitively, this method does not rely on any existing pre-trained foundation models. It simply uses one model, which not only helps to enhance the data diversity (like MAE in DQ), but can also do the dataset split (like bin generalization/selection in DQ).

We conducted extensive experiments on multiple datasets, including ImageNette (a 10-class subset of ImageNet), CUB-200-2011, Food-101, and ImageNet. We evaluated the performance and stability of the deep models (i.e., both ResNet and ViT models) trained on the selected subset using our proposed method under various data keep ratios. The experimental results show that the proposed DQ_V2 can eliminate the drawbacks of DQ's dependence on pre-trained models while achieving performance improvements across multiple datasets.

## 2 RELATED WORK

### 2.1 DATASET QUANTIZATION

Coreset selection is a crucial technique for reducing the computational complexity of deep learning models by selecting a representative subset from large-scale datasets. Many efforts have been made to address this issue, including geometry-based methods (Agarwal et al., 2020; Chen et al., 2012; Sener & Savarese, 2018), uncertainty-based methods (Coleman et al., 2019), error-based methods (Toneva et al., 2019; Paul et al., 2021), decision boundary-based methods (Ducoffe & Precioso, 2018; Margatina et al., 2021), gradient matching-based methods (Mirzasoleiman et al., 2020; Killamsetty et al., 2021), and submodularity-based methods (Iyer et al., 2021). On the other hand, dataset distillation (Sachdeva & McAuley, 2023) is another important technique for compressing large-scale datasets, which aims to synthesize a representative subset from the original dataset.

However, recent research has shown that regardless of coreset selection or dataset distillation, most existing methods face challenges in low scalability to larger datasets (Guo et al., 2022; Zhou et al., 2023). Therefore, Dataset Quantization (DQ), a combination of coreset selection and dataset distillation, has been proposed to address these challenges (Zhou et al., 2023; Zhao et al., 2024). It can effectively select a representative subset from large-scale datasets while maintaining high performance under all data keep ratios. However, DQ-based framework is heavily based on various pre-trained models, that is, Masked AutoEncoder (MAE) (Zhao et al., 2024) and a pre-trained ResNet model. These pre-trained models dominate the computational complexity and stability of the DQ method. Directly removing them leads to performance degradation and increased variance in trained model under different random seeds. In this work, we rethink the necessity of these pre-trained models in the DQ method and propose a new method that achieves efficient coreset selection without relying on any pre-trained models.

## 2.2 Data Augmentation

Data augmentation (Shorten & Khoshgoftaar, 2019) plays an essential role in improving model robustness and generalization ability. Traditional data augmentation methods focus mainly on simple image transformations, such as rotation, flipping, and color adjustment. Recent studies have explored more advanced data augmentation strategies, such as random erasing (Zhong et al., 2020), Mixup (Zhang et al., 2018), CutMix (Yun et al., 2019), and "Copy and paste" (Dwibedi et al., 2017; Ghiasi et al., 2020). These methods have achieved significant success in enhancing the performance and stability of vision models. Although these data augmentation methods have achieved significant success in improving model performance, they generally lack consideration of image semantic structure. Cao & Wu (2022) introduce a novel data augmentation method that leverages the inductive bias of random CNNs to preserve semantic objects while mixing up the background. How to design data augmentation strategies that can both maintain image naturalness and effectively enhance model learning ability remains an open question. In this work, we first observe that the pre-trained MAE model is actually equivalent to a data augmentation method, which introduces prior knowledge and implicit regularization into the training process. Thus, this observation motivates us to explore a new data augmentation strategy that can replace the MAE model in the DQ method.

## 3 Methodology

### 3.1 Preliminaries

Suppose that we have a large dataset $\mathcal{D} = \{(x_i, y_i)\}_{i=1}^T$, where $x_i$ is the $i$-th image and $y_i$ is the corresponding label, and $T$ is the total number of training samples. Coreset selection aims to choose a optimal small subset $D_S$ from a large-scale dataset $\mathcal{D}$, where $D_S \subset \mathcal{D}$ and $|D_S| \ll |\mathcal{D}|$. The model trained on $D_S$ can achieve comparable performance to the model trained on the entire dataset $\mathcal{D}$. Finally, the model trained on the coreset $D_S$ can be used to make predictions on the test set. As discussed above, most coreset selection and dataset distillation methods suffer from some obvious drawbacks, such as poor generalization and low scalability. Therefore, Zhou et al. (2023) proposed a new method, Dataset Quantization (DQ), which consists of three main steps: 1) dataset bin generalization, 2) selection of subset bin , and 3) image pixel quantization. The entire DQ pipeline is shown in Figure 1 (a).

The first step aims to generate multiple non-overlapping dataset subsets (referred to as bins), each containing representative and diverse samples. Where DQ leverages the traditional coreset selection method, i.e., GraphCut method (Iyer et al., 2021) to select the most representative samples. Here, a pre-trained ResNet model is used to extract features for all images, and the GraphCut score is calculated for each unselected sample when added to the current bin. The second step involves random sampling of the generated bins to form the final compressed dataset. This design introduces additional randomness, contributing to improved model robustness and generalization. The final step is to further reduce storage requirements and enhance data quality. This process involves image patching, importance scoring, patch selection, and image reconstruction. Here the pre-trained Masked Autoencoder (MAE) decoder is used to reconstruct the complete image.

Since DQ can achieve state-of-the-art performance on various datasets, especially large-scale datasets like ImageNet, it has attracted significant attention from the research community. Despite

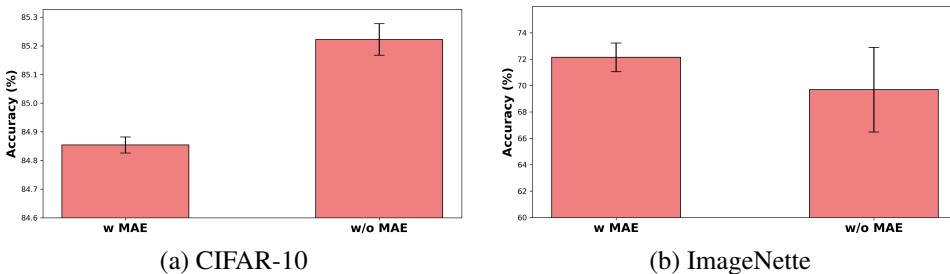

(a) CIFAR-10             (b) ImageNette

Figure 2: The effect of pre-trained MAE model. We compare the performance of DQ and DQ without MAE on CIFAR-10 and ImageNette datasets.

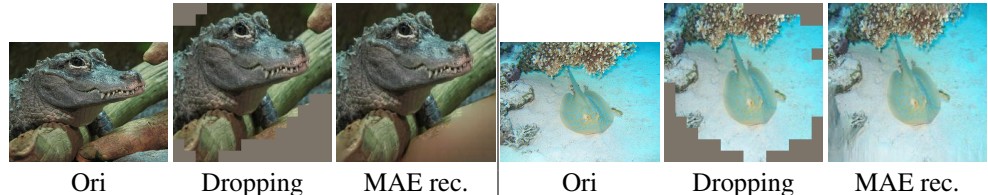

Ori     Dropping     MAE rec.      Ori     Dropping     MAE rec.

Figure 3: The examples of image dropping and MAE reconstruction. The original images are from the ImageNette dataset.

significant advancements in coreset selection, DQ still faces several key challenges, particularly in computational efficiency and method stability. We argue that this potential issue is due to the heavy reliance on large pre-trained models, especially the ViT-Large architecture-based MAE model used in the pixel quantization step. So, in this paper, we study the essential factors of the pre-trained models in the DQ method that affect the performance and stability of the trained model.

## 3.2 THE EFFECT OF PRE-TRAINED MODELS

Our goal is to ablate the role of the pre-trained models in the DQ method. Thus, we conducted a series of controllable experiments to investigate the impact of pre-trained models, particularly the Masked Autoencoder (MAE) model, on the performance and stability of the DQ method. Our results reveal that removing the MAE model significantly degrades DQ's performance and increases the variance in model training.

We remove the pixel quantization step in the original DQ method and directly use the selected images from the second step to train the model. This process is equivalent to removing the pre-trained MAE model in the DQ method. We conducted experiments on both CIFAR-10 and ImageNette datasets with different random seeds, and we report the mean accuracy and variance of the trained models in Figure 2. The results show that the pre-trained MAE model contributes significantly to reducing the variance in model training, which makes the final trained model achieve consistent performance under different random seeds. This also helps researchers reproduce the results and compare different methods more effectively. However, removing the MAE model will not affect the mean accuracy, but the accuracy increases slightly in CIFAR-10. We think this is because the image scale of CIFAR-10 is relatively small, and the MAE model may introduce some noise that affects the model's performance. But, on ImageNette, the MAE model indeed helps to improve the model's performance.

In general, these results reveal an interesting phenomenon: *MAE has a crucial impact on DQ's performance and stability of DQ for larger datasets, while its role is relatively minor for smaller datasets*. Therefore, we further analyze the underlying reasons for this phenomenon.

In the third step of DQ, the training images are first removing the less-informative patches based on the importance scores and then reconstructing the complete image using the pre-trained MAE model. See Figure 3. In fact, the MAE model is pre-trained on a large-scale dataset, which makes it encapsulate extensive image prior knowledge. Since the removing region usually locates in the background or less important regions, the MAE reconstruction actually infers the missing regions

based on the semantic structure of the object in the image. We argue that the MAE reconstruction is equivalent to a data augmentation method which introduces prior knowledge and implicit regularization into the training process. It can also improve the diversity and quality of the training samples, thus improving the performance and stability of the model.

However, this enhancement comes at the cost of increased computational complexity and dependence on large pre-trained models. This raises a crucial question: *Can we design a more efficient method that achieves or surpasses DQ's performance without relying on large pre-trained models?* In the following sections, we will answer this question by introducing a novel data augmentation strategy that can replace the MAE model in the DQ method.

### 3.3 Semantically Aware Data Augmentation

As discussed before, the pixel quantization step mainly preserves the semantic object while dropping the background regions that are less-informative or less important for the model training. However, the reconstruction process not only imputes the missing regions, but also slightly modifies the original image, which can be viewed as a form of data augmentation. This process is similar to the classical data augmentation method, i.e., CutMix (Yun et al., 2019). CutMix generates new training samples by cutting and pasting patches between training images while adjusting the corresponding labels proportionally, enhancing the model's robustness and generalization.

However, CutMix may randomly cut foreground objects, which cannot hold the semantic object. We simply replace CutMix with the MAE reconstruction step in the DQ method and directly apply the composed data augmentation to the DQ pipeline. We find that the results on the ImageNette dataset show that the CutMix method has a significant performance drop compared to the MAE reconstruction method. See the results reported in the following section. Thus, we need to design a more effective data augmentation strategy that not only maintains the information of semantic objects but also does not rely on any pre-trained models.

We investigated two data augmentation strategies in recent work that help to relieve the dependency on the pre-trained models in the DQ method. Tobias leverages the inductive bias of randomly initialized CNNs to preserve semantic objects while mixing up the background. This method can effectively maintain the naturalness of the image and introduce beneficial variations. See Figure 5a. In addition, we mix the Tobias data and the original images to further enhance the diversity of the training data. We also report that using appropriately mixing rates can further improve the model's performance. While this strategy offers the following advantages:

- Semantic Consistency: By preserving the image's main object region, it ensures that augmented images maintain the original semantic information.

- Diversity Introduction: The replacement of background regions introduces new visual contexts, increases data diversity, and improves model generalization.

- Computational Efficiency: Compared to using large pre-trained models (like MAE), this method has lower computational overhead and requires no additional model dependencies, making it suitable for resource-constrained environments.

### 3.4 Our Proposed Framework: DQ_v2

By integrating the Tobias data augmentation strategy into the DQ framework without the MAE model, we addressed the stability issues of the original DQ method while reducing computational complexity. Our improved method includes the following key steps:

**1) Mask Generation**: Use an early-trained model[1] (e.g., ResNet-50) to generate masks for each image in the training set, localizing the regions of the main objects. This step leverages CNN's inductive bias to effectively identify the main objects in the images. **2) Tobias Data Augmentation**: Based on the generated masks, augment the training data. Specifically, we retain the main object part of the image while replacing the original background with randomly selected backgrounds from other images, creating an expanded training set. This step maintains the original image's semantic

---

[1]The early-trained model is the model trained by several epochs (like 10 epochs), which is similar to the scheme in (Paul et al., 2021).

information while introducing new visual contexts. Then, we mix the Tobias data and the original data in an appropriate ratio to build a new training dataset. **3) Dataset Binning**: Use the early-trained model to extract the visual feature and then apply the GraphCut method (Iyer et al., 2021) to split the mixed training set, generating multiple non-overlapping bins. This step ensures that the selected samples are representative and diverse, keeping the core advantages of the DQ method. **4) Bin Sampling**: Randomly select a certain percentage of images from each bin to form the final core set. This random sampling process further increases the diversity of the data, allowing users to flexibly adjust the proportions of the data to suit the different task requirements. **5) Model Training**: Train the model using the selected core set. As the coreset contains both original and augmented images, the model can learn richer and more robust feature representations, enhancing model performance and stability.

The pipeline of our proposed method, named Dataset Quantization V2 (DQ_V2), is illustrated in Figure 1. Through this design, our method not only resolves the original DQ method's dependence on large pre-trained models but also improves performance and efficiency in multiple aspects. Experimental results show that without using the MAE model, our method can achieve or even surpass the performance of the original DQ, providing an efficient and practical solution for core set selection and data compression.

# 4 EXPERIMENTAL RESULTS AND ANALYSIS

## 4.1 EXPERIMENTAL SETUP

**Datasets:** We conducted experiments on multiple datasets, including ImageNet-30 (a 30-class subset of ImageNet), ImageNette (a 10-class subset of ImageNet), CUB-200-2011, and Food-101. These datasets cover a wide range of image classification tasks, enabling us to comprehensively evaluate the performance of our proposed method. In addition, we also conducted experiments on Tiny-ImageNet to further validate the effectiveness of our proposed method compared to the state-of-the-art methods.

**Implementation Details:** We implement our proposed DQ_v2 method using PyTorch and train the models on NVIDIA V100 GPUs. We use the randomly initialized ResNet-50 model as the backbone for Tobias data augmentation. This model is pre-trained on the corresponding full dataset with 10 epochs, and then be used as the feature extractor for selecting the dataset bins. The number of the dataset bins is set to 10 as default. We also use the timm library (Wightman, 2019) for model training in all datasets. We train the ResNet-50 model[2] for 200 epochs for all experiments with a batch size of 128 and an initial learning rate of 0.1 with a one-cycle learning rate scheduler.

## 4.2 MAIN RESULTS

**Stability Analysis.** As discussed before, the pixel quantization step plays an important role in reducing the variance of the trained model. Therefore, in this part, we first investigate the stability of our proposed DQ_v2 method compared to the original DQ method. See Figure 4 (a).

First, we observe that removing the MAE model from the original DQ method significantly increases the variance in the trained model, while the performance also has a significant drop (from 72.14% to 69.69% in ImageNette dataset). Then, compared to the original DQ method, our proposed DQ_v2 method can achieve a comparable variance, while achieving a higher accuracy of 73.80%. This result indicates that our proposed method can effectively address the stability issue of the original DQ method while maintaining high performance. Furthermore, we conduct experiments on two other datasets, such as CUB-200-2011 and Food-101, and the results show that our proposed method can achieve lower variance and higher accuracy compared to the original DQ method. For instance, on CUB-200-2011 dataset, our proposed method achieves a variance of 0.233, significantly lower than DQ's 0.763. In the Food-101 dataset, our proposed method achieves a variance of 0.0745, also lower than DQ's 0.197.

These results underscore the effectiveness of our method in addressing the instability issue of DQ when the pre-trained model is removed. We attribute DQ_v2's stability primarily to the following

---

[2]If not otherwise specified, the accuracy we report is always using ResNet-50 as the backbone model.

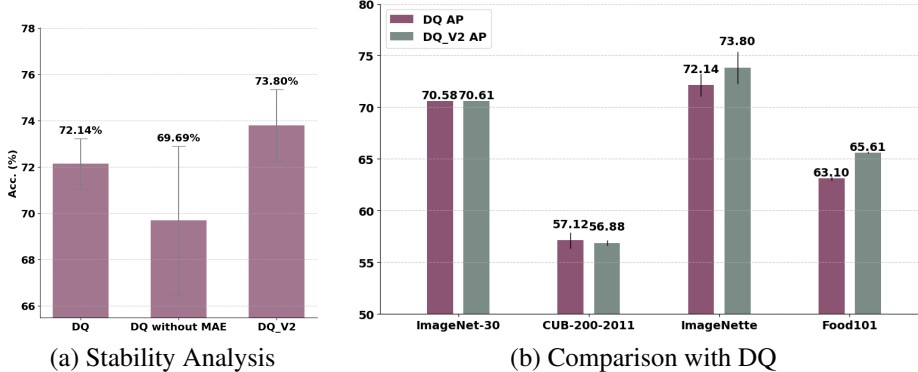

(a) Stability Analysis          (b) Comparison with DQ

Figure 4: Main results of our proposed method. The subfigure (a) shows the stability analysis of our proposed method compared to the original DQ method. The subfigure (b) shows the performance comparison of our proposed method with the original DQ method. We report the mean accuracy (%) and variance in five runs with different seeds.

Table 1: The results of the storage cost. We report the storage cost of the original dataset, the dataset after dropping the less informative region, the dataset after reconstruction, and the dataset generated by Tobias method.

| Method | Original | Dropping | MAE Reconstruction | Ours |
|--------|----------|----------|--------------------|------|
| (GB)   | 3.35     | 0.29     | 0.30               | 1.83 |

factors: 1) By employing semantically-aware background replacement, it provides more diverse training samples, reducing dependence on specific background features while expanding the sample space and mitigating the risk of overfitting. 2) Maintaining a balance of original images and Tobias-augmented images in the dataset preserves the authenticity of the original data while introducing sufficient diversity.

**Comparison with original DQ.** In this part, we mainly evaluate the performance of our DQ_v2 method compared to the original DQ method. Specifically, the results on four representative datasets are shown in Figure 4 (b). The results show that our method achieves comparable or even better performance compared to the original DQ method. For example, our method achieves an average performance gain of 1.57% over four evaluation datasets. especially in the Food-101 dataset, our method achieves a significant performance improvement of 3.98% compared to the original DQ method. Moreover, based on the subset selected by using ResNet-50, we train a ViT model and evaluate the performance on ImageNette. The accuracy of DQ method is 55.30%±2.73%, and ours is 57.67%±1.20%. The results further verify the effectiveness of our proposed method.

In addition, we compare the storage cost of our proposed method with the original DQ method. The DQ method utilizes the patch drop scheme and MAE reconstruction, which claims to reduce the storage cost of the dataset. The results are reported in the Table 1. Therefore, we can see that the pixel Quantization step indeed helps to reduce the storage cost of the training dataset. Although our method needs more storage cost, but we don't need to rely on the pre-trained MAE model, which can reduce the computational complexity and improve the training efficiency. Thus, how to further reduce the storage cost of the training dataset while maintaining the performance of the model is an interesting direction for future research.

**Comparison with State-of-the-arts.**

We mainly compare our proposed DQ_V2 method with the original DQ method and other state-of-the-art coreset selection methods, including random selection (Guo et al., 2022), GraNd (Paul et al., 2021), Grad-Match (Killamsetty et al., 2021), GC (Iyer et al., 2021), etc. Following the default setting in previous work (Zhou et al., 2023; Guo et al., 2022), we conduct the experiments on the Tiny-ImageNet dataset. We used ResNet-50 as the backbone model for all experiments. We vary the data keep ratio from 0.1 to 0.4 and report the performance of the trained models in the test set. The results are shown in the following table: The results show that, the proposed method can achieve the best performance compared to other state-of-the-art methods, especially when the data keep ratio

Table 2: Performance comparison with state-of-the-art methods Tiny-ImageNet datasets. We compare our method with five representative methods. All results of the compared methods refer to (Zhao et al., 2024).

| Ratio/Method | Random | GradNd | GradMatch | GC | DQ | DQ_V2 |
|---|---|---|---|---|---|---|
| 10% | 50.19 | 42.14 | 43.23 | 52.71 | 52.77 | 53.12 |
| 20% | 52.50 | 44.39 | 46.69 | 53.18 | 55.16 | 56.77 |
| 30% | 58.52 | 43.65 | 49.10 | 53.75 | 59.05 | 61.04 |
| 40% | 61.45 | 48.75 | 51.92 | 56.00 | 62.24 | 63.01 |

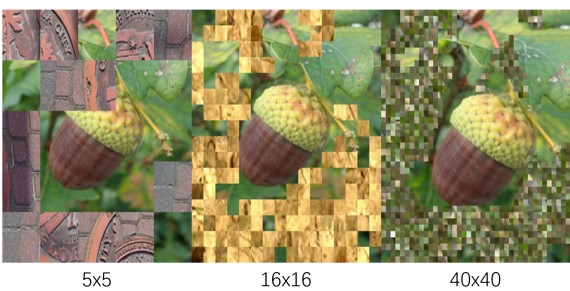

(a) Illustrative examples of different Tobias sizes

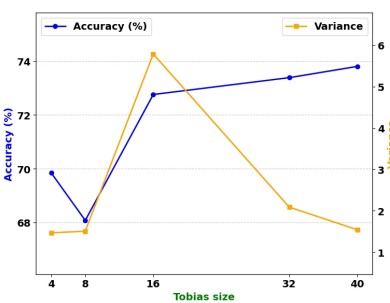

(b) Acc. and Var. for different Tobias sizes

Figure 5: Impact of Tobias size on DQ_v2 performance

is relatively low. While our results are not always competitive when compared to those in (Zhao et al., 2024), we believe that our proposed pipeline can be integrated into this method, where the performance can be further improved.

### 4.3 ABLATION STUDY

In this subsection, we mainly study our method by changing one component while keeping the others fixed. We study the impact of Tobias augmentation, bin split strategy, and mixing ratio between original images and Tobias-generated images on the performance of our proposed method.

**Impact of Tobias Size.** First, we investigate the impact of the Tobias size that is a key parameter in the Tobias data augmentation strategy. The Tobias size determines the granularity of the image patch division. A higher value indicates a finer division, which better preserves the semantic structure. See Figure 5a as an example. We evaluated five different Tobias sizes on the ImageNette dataset, and the results are shown in Figure 5b.

The results show that the Tobias size has a significant impact on the performance of our proposed method. First, we observe that smaller Tobias sizes (e.g., 4×4, 8×8) lead to performance degradation, possibly due to insufficient granularity resulting in incorrect segmentation of the main subject. Second, as Tobias size increases, performance generally shows an upward trend, reaching optimal at 40×40. Finally, larger Tobias sizes (e.g., 32×32, 40×40) not only improve average accuracy but also reduce variance, indicating more stable performance. This phenomenon may stem from more detailed background replacement almost eliminating semantic information in the background, allowing the model to better focus on foreground features while reducing dependence on and overfitting to specific backgrounds. Based on these results, we recommend using a Tobias size of 40×40 for optimal performance and stability.

**Impact of Bin Division Algorithms.**

Then, we analyze the impact of different bin split algorithms on the performance of our proposed method. Similarly to (Zhou et al., 2023), we compare the performance of our method with four different group split strategies, including the GraphCut, Random, Uniform, and EarlyTrain methods, in the ImageNet-30 dataset. The results are shown in Table 3. The results show that the GraphCut method achieves the best performance compared to other bin split methods. This indicates that the GraphCut method can effectively select the most representative samples from the dataset, which can

Table 3: Performance comparison of different bin split algorithms on ImageNet-30

| Method | ImageNet-30 Accuracy |
|---|---|
| DQ_v2 with GraphCut (Iyer et al., 2021) | 70.61% |
| DQ_v2 with EarlyTrain (Paul et al., 2021) | 70.27% |
| DQ_v2 with Random bin generation | 70.29% |
| DQ_v2 with Uniform bin generation | 70.20% |

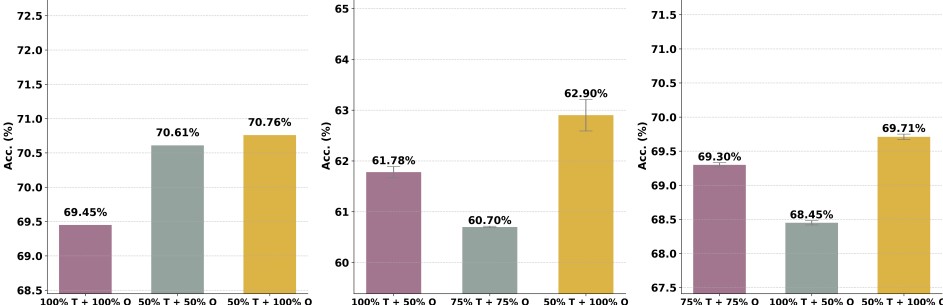

Figure 6: Performance with different T and O ratios in ImageNet-30 (left), CUB-200 (center), and Food-101 (right). We report the mean accuracy (%) and variance in five runs with different seeds.

improve the performance of the trained model. Moreover, the GraphCut method also achieves the lowest variance compared to other methods, which indicates that the GraphCut method can provide more stable performance under different random seeds. We also observe similar results on other datasets. For example, on the Imagenette dataset, the GraphCut method achieves an accuracy of $72.136\% \pm 1.092\%$, while the Uniform method achieves $71.46\% \pm 3.759\%$.

In summary, the GraphCut algorithm not only consistently outperforms Random and Uniform methods across different datasets but also significantly reduces the variance between different random seeds, indicating more stable performance. Therefore, in practice, we recommend using the Graph-Cut method to achieve optimal performance and stability.

**Impact of Mixing Ratio.** Finally, we investigate the impact of the mixing ratio between the original images and Tobias-generated images on the performance of our proposed method. Since we do not need to rely on the pre-trained MAE model, we can simply mix Tobias data and original images to improve the diversity and quality of the data. Thus, we evaluated the performance of our proposed method with different mixing ratios in the ImageNet-30, CUB-200, and Food-101 datasets. The results are shown in Figure 6. We observe that performance consistently achieves the best score when we use all original images together with 50% Tobias-generated images. However, this strategy actually increases the scale of the training dataset. To balance performance and efficiency, we suggest splitting the training dataset into more bins, each maintaining the same scale as that in DQ's setting, to be used for training the model.

## 5 CONCLUSION

In this paper, we introduced DQ_V2, an enhanced version of the Dataset Quantization (DQ) method, which integrates the Tobias data augmentation strategy to address the limitations of the original DQ method. Our proposed framework eliminates the dependency on large pre-trained models, thereby reducing computational complexity and improving training efficiency. Through extensive experiments on multiple datasets, we demonstrated that DQ_V2 achieves superior performance and stability compared to the original DQ method and other state-of-the-art coreset selection techniques.

Our findings suggest that intelligent data augmentation and selection strategies can significantly enhance model performance without relying on extensive pre-training. Future research directions include exploring more advanced semantic-aware data augmentation techniques, extending DQ_V2 to other visual tasks, and developing adaptive parameter adjustment strategies to further improve its flexibility and applicability.

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

## A  THE TOBIAS DATA AUGMENTATION STRATEGY

The Tobias data augmentation strategy includes the following steps:

1. Mask Generation: Use a randomly initialized or lightweight pre-trained CNN to generate masks for each image, identifying the main object regions.

2. Image Segmentation: Divide the image into fixed-size patches.

3. Background Replacement: Based on the generated mask, the patches are retained in the main object region and the background patches are replaced with those of other images.

## B  DISCUSSION AND FUTURE PROSPECTS

Based on our experiments and analysis, this paper summarizes the following key findings.

- **No reliance on large pre-trained models**: DQ_v2 achieves comparable or superior performance to DQ without relying on large pre-trained models.

- **Computational resources and implementation simplicity**: DQ_v2 requires fewer computational resources and is simpler to implement.

- **Excellent performance across multiple datasets and scales**: DQ_v2 performs well on various datasets with different image sizes.

- **Significant improvement in training stability**: DQ_v2 significantly improves the stability of model training, reducing performance variations between different random seeds.

- **Good complementarity with data augmentation**: DQ_v2 demonstrates good complementarity with other data augmentation methods (e.g., Mixup).

Based on these observations, we offer insight into several important questions, with the aim of inspiring the community to rethink core set selection and data augmentation methods.

### B.1  ARE LARGE PRE-TRAINED MODELS NECESSARY?

No. The prior knowledge provided by large pre-trained models is not absolutely necessary. Our experiments show that DQ_v2 can achieve or even surpass DQ's performance without using large pre-trained models like MAE. As mentioned in (He et al., 2019), the prior knowledge from pre-trained models does not necessarily improve model performance on target data. Through simple processing of target data, DQ_v2 can achieve effects similar to using expensive pre-trained large models.

### B.2  IS THE DQ METHOD USEFUL?

Yes. DQ, as a core set selection method, provides a crucial auxiliary tool for the computer vision community. Not only do it saves computational resources required for large-scale data, but also it shortens research cycles, making it easier to obtain encouraging results. We believe these advantages will continue to make DQ play an important role in computer vision research. However, DQ_v2's success indicates that there is still room for further improvement in efficiency and performance through smarter data processing methods.

### B.3  DO WE NEED BIG DATA?

Yes, but the cost of data collection and cleaning needs to be balanced. Although large-scale datasets have advantages in some tasks, general large-scale classification pre-training sets are not always ideal, for example, the cost of collecting ImageNet is largely overlooked. If the benefits of large-scale classification-level pre-training show diminishing returns, collecting data in the target domain might be more efficient. The success of DQ_v2 demonstrates that existing data can be used more effectively through intelligent data augmentation and selection strategies to improve model performance.

### B.4 FUTURE PROSPECTS

DQ_v2 opens new research directions in core set selection and data augmentation. Future research may include the following.

- Exploring more precise and intelligent semantic-aware data augmentation strategies: Developing more advanced data augmentation methods to further enhance model generalization ability and performance.
- Extending DQ_v2 to more visual tasks: For example, object detection and semantic segmentation, to verify its effectiveness in more complex tasks.
- Developing adaptive parameter adjustment strategies: Enabling DQ_v2 to better adapt to different datasets and task requirements, achieving higher flexibility and applicability.

## C   EVALUATION ON DATA AUGMENTATION METHODS.

Since we leverage the Tobias data augmentation strategy to simulate the pixel quantization step, we further investigate the effects of data augmentation methods on the performance of our proposed method. During the final model training, we further add the Mixup and CutMix data augmentation methods to our pipeline, and evaluate the performance of the trained model on the ImageNet-30 dataset.

| Method | DQ_v2 | +Mixup | +CutMix | +Both |
|---|---|---|---|---|
| Accuracy | 70.76 | 73.08 | 71.11 | 73.87 |

As can be seen in the table, traditional data augmentation methods (such as Mixup and CutMix) significantly improve the performance. This indicates that our method is not mutually exclusive with these methods, but can work synergistically.

These observations reveal the following points:

- **Complementarity**: Ours provides a unique form of data augmentation through semantically aware background replacement, which complements well with methods like Mixup (linear interpolation) or CutMix (region replacement). This implies that our method can work in conjunction with other data augmentation techniques to further enhance model performance.
- **Intrinsic Regularization**: Ours may already have achieved a degree of data augmentation effect, resulting in limited additional performance gains when combined with other methods. This suggests that ours has inherent advantages in improving model generalization, reducing dependence on other regularization methods.

## D   EVALUATION OF DIFFERENT BACKBONE.

In the proposed method, we use ResNet-50 as the backbone model for the Tobias data enhancement strategy and also split / select the data sets. So, we evaluated the performance of our proposed method with different backbone models, including ResNet-50, ViT-tiny, RepVGG, and Inception V3. The results are shown in the following table: These results show a similar tendency to the

Table 4: Performance comparison of different backbone models on ImageNete

| Method | Accuracy |
|---|---|
| DQ_v2 with ResNet-50 | $73.80\% \pm 1.54$ |
| DQ_v2 with ViT-tiny | $69.89\% \pm 0.16$ |
| DQ_v2 with RepVGG | $69.95\% \pm 1.17$ |
| DQ_v2 with Inception V3 | $70.57\% \pm 0.27$ |

original Tobias work in (Cao & Wu, 2022), that the ResNet-50 model achieves the best performance

compared to other backbone models. However, we also observe an intriguing phenomenon that the ViT-tiny and Inception models can achieve a lower variance compared to the ResNet-50 model. This indicates that the backbone model can also affect the stability of the trained model. Although the current results are not good enough, we believe that the backbone model can be further optimized to improve the performance and stability of the trained model.

