# OpenReview forum: "Rethinking Dataset Quantization: Efficient Core Set Selection via Semantically-Aware Data Augmentation"
_ICLR.cc/2025/Conference — Submitted to ICLR 2025_

### Official Review · Reviewer_uDVp · 2024-10-25

**Soundness:** 3
**Presentation:** 3
**Contribution:** 2
**Rating:** 5
**Confidence:** 4

**Summary:**

This paper proposes Dataset Quantization V2 (DQ V2), an enhanced version of the original Dataset Quantization (DQ) method, focusing on efficient coreset selection without relying on large pre-trained models like MAE. Instead, DQ V2 integrates a new data augmentation strategy called Tobias, which uses randomly initialized CNNs to preserve the semantic regions of images while replacing background areas, mimicking the effect of pixel quantization. Extensive experiments demonstrate that DQ V2 achieves improved performance and training stability across multiple datasets, while also reducing computational complexity. The results suggest that DQ V2 provides a practical solution for data compression and coreset selection, paving the way for further enhancements in semantic-aware data augmentation and broader applications in complex visual tasks.

**Strengths:**

- The overall writing of the paper is smooth and easy to understand.
- DQ V2 replaces MAE-based quantization with a simple augmentation strategy, achieving better performance without pre-trained models.

**Weaknesses:**

- The paper claims good scalability for the proposed method, but the experiments are still focused on smaller datasets and do not include evaluations on mainstream large-scale datasets like ImageNet-1k.
- The coreset selection methods chosen for comparison, such as GraNd, Grad-Match, and GC, are from 2021. The paper should include comparisons with more recent coreset selection and dataset quantization methods.

**Questions:**

The goal of DQ is to reduce training data volume and improve data efficiency. Since the proposed method uses data augmentation, does it significantly increase the dataset size, potentially resulting in similar training costs as regular training?

---

> ### Author Response · Authors · 2024-12-04
>
> We sincerely appreciate the reviewer's thoughtful analysis. Below we address each concern in detail:
>
> 1 Regarding Large-scale Dataset Evaluation:
> We acknowledge the limitation of not evaluating on ImageNet-1k. This was primarily due to computational constraints, as our research was conducted using a single RTX 4090 GPU. While this limitation affects the direct comparability with some previous works, our results on multiple smaller-scale datasets demonstrate the effectiveness of our approach. The consistent performance improvements across diverse datasets suggest potential scalability to larger datasets.
>
> 2 Regarding Recent Baseline Comparisons:
> We sincerely acknowledge that not including recent state-of-the-art coreset selection and dataset quantization methods is a limitation of our work. Our initial comparison focused primarily on DQ as we considered it the state-of-the-art approach at that time and mistakenly assumed it had encompassed comparisons with most existing methods. While our method demonstrates advantages in computational efficiency and improved performance compared to DQ, we recognize that a more comprehensive evaluation including recent methods would have provided a more complete positioning of our approach within the current landscape of coreset selection techniques.
>
> 3 Regarding Training Costs and Dataset Size:
> We appreciate this concern about potential training cost increases. However, we would like to clarify that while our method employs data augmentation, it does not increase the final training data volume. Similar to the original DQ's use of MAE for data enhancement, we employ CNN-based augmentation but maintain the same final dataset size. Specifically, we create a mixed dataset by combining 50% original images and 50% augmented images, ensuring the total size remains consistent with the original DQ method's selection ratio. Therefore, the training costs are comparable to other coreset selection methods as we maintain the same proportion of training data. This approach enhances data diversity while maintaining computational efficiency.
>
> Our method aims to provide a more efficient alternative to DQ by eliminating the need for large pre-trained models while preserving its core advantages. The identified limitations in our evaluation framework provide valuable direction for future work, where we can more comprehensively assess our method against the full spectrum of current approaches in coreset selection.

---

### Official Review · Reviewer_qJFV · 2024-10-29

**Soundness:** 2
**Presentation:** 2
**Contribution:** 3
**Rating:** 5
**Confidence:** 5

**Summary:**

This work proposes DQ_v2, a corset selection method. To remove the pre-trained MAE in DQ, the authors investigate a data augmentation scheme, which can simulate the steps of pixel compression and reconstruction in DQ. Finally, the authors show the performance on several benchmark datasets, including CUB-200, Food-101, and ImageNet. The idea of using data augmentation to replace pre-trained MAE in DQ is somewhat novel to me. However, some critical concerns remain, please see weakness.

**Strengths:**

1. Using semantical-aware data augmentation to remove the pre-trained MAE model in DQ is interesting.
2. The paper is well-organized.
3. Experimental results show that the proposed DQ_v2 eliminates the drawbacks of DQ's dependence on pre-trained.
4. The proposed method achieves performance improvement on multiple datasets.

**Weaknesses:**

1. In line 278, the authors say that the corset contains both original and augmented images. However, as far as I know, most existing corset selections only select original images from the datasets, meaning that there are no augmented images in corsets. So is this a fair comparison between DQ_v2 and other corset selection methods?
2. The literature review section lacks comprehensiveness. Numerous recent studies closely related to the topic have not been studied, such as [1-5], which may affect the context and clarity of the proposed approach.
[1] Tan, Haoru, et al. "Data pruning via moving-one-sample-out." Advances in Neural Information Processing Systems 36 (2024).
[2] Xia, Xiaobo, et al. "Moderate coreset: A universal method of data selection for real-world data-efficient deep learning." The Eleventh International Conference on Learning Representations. 2022.
[3] Yang, Shuo, et al. "Dataset pruning: Reducing training data by examining generalization influence." arXiv preprint arXiv:2205.09329 (2022).
[4] Maharana, Adyasha, Prateek Yadav, and Mohit Bansal. "D2 pruning: Message passing for balancing diversity and difficulty in data pruning." arXiv preprint arXiv:2310.07931 (2023).
[5] Yang, Suorong, et al. "Not All Data Matters: An End-to-End Adaptive Dataset Pruning Framework for Enhancing Model Performance and Efficiency." arXiv preprint arXiv:2312.05599 (2023).
3. In the semantic data augmentation section, the authors enhance diversity by replacing image backgrounds. However, it’s unclear if the potential for semantic ambiguity was considered—for instance, whether the new backgrounds might inadvertently introduce other objects, which could affect the intended semantics.
4. The authors report only storage costs, but I recommend adding a comparison of training costs as well. This would provide a more comprehensive assessment of the method’s efficiency and practical applicability.
5. The practical significance of the proposed method is unconvincing due to limited experimental validation. In the experimental section, all benchmark comparisons are with methods published before 2021. The compared baselines are outdated. While authors claim the comparison with state-of-the-art, many existing SOTA methods [1-5] are not compared. This weakens the method’s practical performance and significance.

**Questions:**

Please see weakness.

---

> ### Author Response · Authors · 2024-12-04
>
> We sincerely appreciate the reviewer's thoughtful analysis. Below we address each concern in detail:
>
> 1 Regarding Fair Comparison with Coreset Selection Methods:
> We believe our comparison is fair for two key reasons. First, while we employ data augmentation, we maintain dataset size parity during the bin creation and selection process by mixing original and augmented images in equal proportions. This ensures that the total number of images participating in coreset selection remains identical to the original dataset size. Second, regarding data regularization, it's worth noting that the original DQ method also employs implicit regularization through its MAE model, which was pretrained on ImageNet-1K. In contrast, our method uses a randomly initialized model, avoiding the introduction of any external knowledge. This actually makes our approach more "pure" in terms of only learning from the target dataset.
>
> 2 Regarding Literature Review and Baselines:
> We sincerely acknowledge that not including recent important works (Tan et al., 2024; Xia et al., 2022; Yang et al., 2022; Maharana et al., 2023; Yang et al., 2023) is a limitation of our work. Our initial comparison focused primarily on DQ as we considered it the state-of-the-art approach at that time and mistakenly assumed it had encompassed comparisons with most existing methods. While our method demonstrates advantages in computational efficiency and improved performance compared to DQ, we recognize that a more comprehensive evaluation including these recent methods would have provided a more complete positioning of our approach within the current landscape of coreset selection techniques.
>
> 3 Regarding Semantic Ambiguity in Background Replacement:
> We appreciate this concern about potential semantic ambiguity. However, our method is specifically designed to avoid introducing unintended semantic information. As demonstrated in Figure 5a, the replacement backgrounds consist of randomly shuffled patches that contain only texture information, not semantic content. These randomized, small-scale patches, when combined, are highly unlikely to form recognizable objects or introduce semantic ambiguity. The background replacement process preserves the semantic integrity of the main object while only introducing variation in non-semantic texture patterns.
>
> 4 Regarding Training Cost Analysis:
> We acknowledge the limitation in our quantitative analysis of computational costs. While our method inherently requires less computation by eliminating the need for a large pretrained model (replacing ViT-Large with ResNet-18), we agree that explicit measurements of training costs and coreset selection computational overhead would have strengthened our claims. This would indeed provide a more comprehensive assessment of our method's practical efficiency.
>
> 5 Regarding Experimental Validation:
> As mentioned in our response to the literature review concern, we acknowledge that our experimental comparisons could have been more comprehensive by including recent state-of-the-art methods. While our focus on comparing with DQ was motivated by its significance as a recent breakthrough in coreset selection, we recognize that including comparisons with the mentioned recent works would provide a more complete evaluation of our method's practical significance.
>
> These limitations in our evaluation framework provide valuable direction for future work, where we can more comprehensively assess our method against the full spectrum of current approaches in coreset selection.

---

### Official Review · Reviewer_kJYW · 2024-10-29

**Soundness:** 2
**Presentation:** 2
**Contribution:** 2
**Rating:** 5
**Confidence:** 4

**Summary:**

This paper examines the limitations of the DQ method and proposes corresponding improvements. The authors believe that using a pretrained MAE in DQ may cause issues, so they conducted experiments to see the impact on DQ when MAE is removed. The experiments, in a way, demonstrate the importance of MAE. The authors suggest using Tobias data augmentation as a substitute for MAE. According to their results, it is possible to achieve accuracy comparable to or even better than the previous DQ without using MAE.

**Strengths:**

1. The method proposed by the authors does indeed achieve comparable or even higher results without using MAE.

2. The authors conducted extensive ablation studies on the parameters of the method itself, including experiments on patch size and data selection methods.

**Weaknesses:**

1. The motivation of this paper is somewhat unclear. From my understanding, the main value of DQ lies in reducing dataset size and storage requirements. However, as shown in Table 1, this method actually increases the storage usage of DQ. The problem it addresses is the need for a pretrained MAE in the original DQ, yet the authors' experiments do not highlight any obvious issues caused by using MAE. In my view, the authors have optimized a relatively minor aspect while losing sight of one of DQ’s key contributions. It would be beneficial for the authors to further elaborate on the advantages of this method.

2. The logic of the proposed method is unclear. The authors first apply Tobias data augmentation, followed by dataset selection—what is the advantage of this sequence? What would the outcome be if Tobias data augmentation were added directly at the end based on DQ?

3. The conclusions regarding line 210 may have some bias, as MAE was pretrained on ImageNet, which likely results in better reconstruction performance on ImageNette. The variables here are not limited to dataset size, so the effectiveness may not necessarily be due to the dataset size alone. It could also be influenced by the effectiveness of MAE itself.

**Questions:**

The biggest question is what specific negative effects MAE actually introduces, as the authors' experiments and analysis do not clearly convey any significant drawbacks to using MAE.

---

> ### Author Response · Authors · 2024-12-04
>
> We sincerely appreciate the reviewer's thoughtful analysis. Below we address each concern in detail:
>
> 1 Regarding Motivation and Core Contributions:
> We respectfully disagree with the characterization that storage reduction is DQ's primary contribution. The main value of DQ lies in its computational efficiency while maintaining high performance in coreset selection. While our method does require more storage (as shown in Table 1), it significantly reduces computational complexity by eliminating the need for the ViT-Large architecture used in MAE, which is the most computationally intensive component of DQ. In today's computing landscape, where storage is relatively inexpensive and even large-scale datasets like ImageNet-1K only require around 100GB of storage space, the computational cost far outweighs storage concerns. Our method preserves DQ's advantages while further reducing computational overhead, making coreset selection more accessible and efficient.
>
> 2 Regarding Method Logic and Design Choices:
> The sequence of applying Tobias data augmentation before dataset selection is a deliberate design choice. This approach allows the coreset selection algorithm to consider both augmented and original images simultaneously when identifying the most representative subset. If we were to perform coreset selection first or apply Tobias augmentation after DQ, the selection process would not account for the augmented samples, potentially leading to less representative core sets. Our approach ensures that the selection algorithm can optimize across both original and augmented images to identify the most effective subset.
>
> 3 Regarding Dataset Size Analysis and MAE Performance:
> We acknowledge that our discussion around line 210 could have been more precise. Our intention was to highlight that MAE's effectiveness varies with image resolution rather than dataset size per se. For larger resolution images (224×224), MAE can leverage more information during reconstruction, potentially providing more effective data augmentation and implicit regularization. Conversely, for smaller resolution datasets like CIFAR, MAE's reconstruction capabilities may be limited by insufficient information. This observation is supported by MAE's strong performance on Food-101, which also features high-resolution images. We acknowledge the reviewer's astute observation that MAE's pretraining on ImageNet-1K (He et al., 2022) might contribute to its effectiveness on ImageNette. However, we note that the strong performance on Food-101 suggests the benefits extend beyond just ImageNet similarity.
>
> Regarding the specific negative effects of MAE:
> There are several important considerations. First, the computational overhead of MAE, which uses a ViT-Large architecture, is substantial and arguably unnecessary. This is particularly problematic since coreset selection methods are typically employed in resource-constrained scenarios, where running such large models may be impractical or inefficient.
> Second, and perhaps more fundamentally, the use of an ImageNet-1K pretrained MAE model introduces implicit prior knowledge into the process. This pretraining effectively incorporates external information and regularization priors from ImageNet-1K into the coreset selection process. Consequently, when evaluating DQ against other coreset selection methods, the comparison isn't entirely fair - it's unclear whether DQ's performance stems from the training data alone or is significantly influenced by the implicit ImageNet-1K knowledge encoded in the MAE model. We believe this might be one of the key reasons behind DQ's remarkable performance on ImageNet-1K.
> While DQ's authors presented MAE primarily as a means for storage reduction, our analysis suggests that the MAE model might be serving another crucial role - providing implicit regularization through its pretrained knowledge. This hypothesis is supported by our experimental results comparing performance with and without MAE across different datasets. Our method achieves comparable or better performance while avoiding these potential concerns, offering a more transparent and computationally efficient approach that relies solely on the target dataset.
>
>
>
> [Reference]
> He, K., Chen, X., Xie, S., Li, Y., Dollár, P., & Girshick, R. (2022). Masked autoencoders are scalable vision learners. In Proceedings of the IEEE/CVF conference on computer vision and pattern recognition (pp. 16000-16009).

---

### Official Review · Reviewer_QWVA · 2024-11-04

**Soundness:** 2
**Presentation:** 3
**Contribution:** 2
**Rating:** 5
**Confidence:** 4

**Summary:**

This paper addresses the high computational cost of Dataset Quantization (DQ) due to its reliance on large pre-trained models like MAE and ResNet. They propose DQ V2, which removes pre-trained models by using a random CNN-based data augmentation that retains semantic structure by masking objects and replacing backgrounds, enhancing diversity without costly models. The goal of data augmentation (synthesizing) in their pipeline is to enhance data diversity and representation without relying on costly pre-trained models.

Evaluation: Evaluated on ImageNette, CUB-200-2011, Food-101, and ImageNet-30, DQ v2’s performance is compared with DQ’s. DQ v2 achieves comparable or better performance than the original DQ method, showing an average improvement of about 1.57%.

**Strengths:**

1. Computational Efficiency: By removing the reliance on large pre-trained models, DQ V2 lowers computational costs.

2. Good insight for data augmentation: The pre-trained MAE model is equivalent to a data augmentation method (in introducing prior knowledge and implicit regularization into the training process)

3. The writing is clear and easy to follow.

**Weaknesses:**

1. Lack of Quantitative Analysis on Computational Gains: While the paper claims computational benefits from replacing the MAE model with a CNN-based data augmentation strategy, it lacks specific measurements or comparisons to substantiate these gains. A quantitative analysis—such as GPU hours, memory usage, or training time—would provide stronger evidence of the efficiency improvements in DQ V2.

2. Missing Baselines: I noticed that some recent coreset selection baselines for deep learning are missing: D2 Pruning[1], CCS[2], Moderate[3]. Those baselines seem to have a stronger performance than the proposed methods.

3. Missing evaluation on ImageNet-1k: the paper argues that DQ-V2 is more efficient than DQ, but the method is only evaluated on the ImageNet subset. Previous methods including DQ all conducted evaluation on ImageNet-1k. It will be good to include an ImageNet-1k evaluation to demonstrate the scalability of the proposed methods.

4. The data augmentation part is confusing: the goal of data quantization and coreset selection is to reduce the size of the training dataset, but the data augmentation method proposed in the paper expands the datasets -- the final expanded training dataset can be even larger, which is contradicted to the goal of coreset selection.

5. Ablation study on data augmentation: The paper would benefit from a more detailed ablation study to assess the effectiveness of the data augmentation method used in DQ V2. Testing different data augmentation configurations (e.g., no augmentation, alternate augmentation techniques) would clarify its impact and help refine the methodology.

[1] Maharana, Adyasha, Prateek Yadav, and Mohit Bansal. "D2 pruning: Message passing for balancing diversity and difficulty in data pruning." ICLR 2024

[2] Zheng, Haizhong, Rui Liu, Fan Lai, and Atul Prakash. "Coverage-centric coreset selection for high pruning rates." ICLR 2023

[3] Xia, Xiaobo, Jiale Liu, Jun Yu, Xu Shen, Bo Han, and Tongliang Liu. "Moderate coreset: A universal method of data selection for real-world data-efficient deep learning."  ICLR 2023

**Questions:**

See weakness

---

> ### Author Response · Authors · 2024-12-04
>
> We sincerely appreciate the reviewer's thorough evaluation and constructive feedback. Below, we address each concern raised:
>
> 1. Regarding Computational Efficiency Analysis:
> We acknowledge the limitation in our quantitative analysis of computational gains. While our method utilizes ResNet-18 instead of ViT-large (as used in MAE), which inherently suggests significant computational savings due to the substantial architectural differences (ResNet-18: ~11M parameters vs. ViT-large: ~307M parameters), we agree that explicit measurements would have strengthened our claims. This architectural efficiency stems from eliminating the need for a large-scale pre-trained model while maintaining comparable performance.
>
> 2. Regarding Missing Baselines:
> We acknowledge that not including recent important baselines (D2 Pruning, CCS, and Moderate) is a limitation of our work. Our initial comparison focused primarily on DQ as we considered it the state-of-the-art approach at that time and mistakenly assumed it had encompassed comparisons with most existing methods. This was an oversight on our part. While our method demonstrates advantages in computational efficiency and improved performance compared to DQ, we recognize that a more comprehensive evaluation including these recent methods would have provided a more complete positioning of our approach within the current landscape of coreset selection techniques. This comparison would be valuable for the community to better understand the relative strengths and trade-offs of different approaches.
>
> 3. Regarding ImageNet-1k Evaluation:
> We acknowledge the limitation of not evaluating on ImageNet-1k. This was primarily due to computational constraints, as our research was conducted using a single RTX 4090 GPU. While this limitation affects the direct comparability with some previous works, our results on multiple smaller-scale datasets demonstrate the effectiveness of our approach. We believe the consistent performance improvements across diverse datasets suggest potential scalability to larger datasets.
>
> 4. Regarding Data Augmentation Strategy:
> We appreciate the opportunity to clarify our data augmentation approach. We would like to emphasize that our method does not actually increase the dataset size. Similar to the original DQ's use of MAE for data enhancement, we employ CNN-based augmentation but maintain the same final dataset size. Specifically, we create a mixed dataset by combining 50% original images and 50% augmented images, ensuring the total size remains consistent with the original DQ method's selection ratio. This approach enhances data diversity while maintaining computational efficiency.
>
> 5. Regarding Ablation Studies:
> We acknowledge that a more comprehensive ablation study would strengthen our work. While we did conduct experiments without augmentation that showed significant performance degradation compared to the original DQ, we agree that examining various augmentation configurations would provide valuable insights. This observation led us to understand that MAE in the original DQ effectively functions as a data augmentation method, introducing beneficial prior knowledge and implicit regularization into the training process.
>
> We appreciate the reviewer's careful evaluation and believe addressing these points would strengthen our work. While the submission period has concluded, these insights will be valuable for future research directions in this area.

---

### Meta-Review · Area_Chair_UB7D · 2024-12-16

**Metareview:**

This paper proposes a novel coreset selection method based on dataset quantization, where rather than relying on a pretrained MAE  the authors rely on data augmentation using a randomly initialized ResNet CNN.
Strengths mentioned in the reviews include the computational efficiency of the method, its insights regarding MAE and data augmentation, and the clear presentation.
Weaknesses include: the experimental execution (quantitative analysis of computational gains, missing baselines, lack of experiments on ImageNet), comparison to methods not using data augmentation, and limited comparison to recent methods.

**Additional Comments On Reviewer Discussion:**

The authors provided a rebuttal in response to the reviews. The rebuttal did address some points of the reviewers (eg about data augmentation in the context of coreset selection), but not others such as comparison to more recent related work. The reviewers (except one) acknowledge the rebuttal, but maintain a negative recommendation. I'm following this unanimous recommendation.

---

### Decision · Program_Chairs · 2025-01-22

Reject